# Development, Testing and Characterization of Al NanoTiC$_p$ Composites through Powder Metallurgy Techniques

**Gaurav Bajpai** [1,*], **Anuradha Tiwari** [2], **Rajesh Purohit** [3,*], **Vijay Panchore** [3], **Rashmi Dwivedi** [4]
**and Kosaraju Satyanarayana** [5]

1   Department of Mechanical Engineering, Indian Institute of Technology, Kanpur 208016, India
2   Department of Physics, Brahmanand Degree College, Chhatrapati Shahu Ji Maharaj University,
    Kanpur 208024, India; tiwarianuradha068@gmail.com
3   Department of Mechanical Engineering, Maulana Azad National Institute of Technology,
    Bhopal 462003, India; vijaypanchore36@gmail.com
4   Department of Mechanical Engineering, Sagar Group of Institutions—SISTec, Bhopal 462036, India;
    rashmidwivedi29@gmail.com
5   Department of Mechanical Engineering, Gokaraju Rangaraju Institute of Engineering and Technology,
    Hyderabad 500090, India; satya.kosaraju@griet.ac.in
*   Correspondence: bajpai2008gaurav@gmail.com (G.B.); rpurohit73@gmail.com (R.P.)

**Abstract:** In the present scenario, weight diminution and strength enrichment are the main requirements for escalating the application of a nano composite material in different sectors. Several industrial sectors, such as automobile, defense and aerospace, are making various components of nano composites with the help of powder metallurgy processing. In this study, Al nanoTiC$_p$ composites (2, 4 and 6 wt %) were contrived through modified powder metallurgy (PM) techniques with the help of Cold Isostatic Compaction process (CIP). The mechanical properties such as density, porosity, micro-hardness, compressive strength and indirect tensile strength were increasing with the reinforcement of nanoTiC$_p$ particles up to 4 wt % in Al metal matrix composites. Nevertheless, clustering of nanoTiC$_p$ particles were found at 6 wt %, which is also observed in SEM images.

**Keywords:** powder metallurgy (PM); cold isostatic compaction process (CIP); Al nanoTiC$_p$ composites; clustering; mechanical properties

## 1. Introduction

In the 21st Century, the application of nanotechnology has rapidly enhanced in all areas. There is a big scope in the field of nano composites in the area of material science because the function of nano composite materials enhances several properties such as chemical, mechanical, optical and/or physical properties in different aspects with light weight [1]. Nowadays, scientists have a big interest in aluminum matrix nano composites (AMNCs) due to their high stiffness, lightweight, high strength-to-weight ratio, lowercost, ease of production and high dimensional tolerance [1]. It is widely used in automobile sectors and the aerospace industry [2]. Nevertheless, overall strength is found less in aluminum, therefore, to enhance this strength; nano particles are reinforced in aluminum matrix nano composites. It is observed that this overall strength is totally dependent of particle size spacing, volume fraction and the nature of matrix and the reinforcement of the interface [3]. A lot of literature is available about several ceramic nano particles, such as SiC, B$_4$C, h-BN, TiC$_p$ and MgO that are reinforced in the AMCs to enhance the wear properties, mechanical properties and microstructure of the composites [4–12]. Nevertheless, TiC$_p$ is a unique ceramic that shows high hardness, low wear, relatively high thermal stability and good wet ability due to its controlled three-dimensional structures [13]. Titanium Carbide (TiC$_p$) is a good ceramic that is used in wear-resistant coating on tool bits, watch machinery and space craft [14]. When TiC$_p$ is added to an Al metal matrix, it enhances the properties of the aluminum metal matrix without an increment in the weight of the Al

metal matrix [15]. Various research reports are available on the thermal properties, wear behavior and material characterization of the Al nanoTiC$_p$ composites.

In this study, Al nanoTiC$_p$ composites have been fabricated by the solid-state sintering process, i.e., powder metallurgy. Different wt % of TiC$_p$ nano particles (2, 4 and 6 wt %) were used for the synthesis of nano composites. The mechanical properties such as micro-hardness, porosity, tensile strength, compressive strength and indirect tensile strength have been investigated.

## 2. Materials Used in the Present Study

Aluminum powder (APS 30–50 μm, Purity: 99%) and nanoTiC$_p$ powder (APS 50–80 nm, Purity: 99.9%) were purchased from BVS Enterprises, Mandideep Bhopal. To avoid difficult determination of critical temperature for infiltration, problems due to fluidity or wettability at matrix-reinforcement interface as well as harmful reactions at the interface in stir casting, the powder metallurgy process with cold isostatic compaction was used in the present study.

## 3. Fabrication of Al NanoTiC$_p$ Composites through Powder Metallurgy Process

The modified powder metallurgy route is followed to prepare Al nanoTiC$_p$ composites with 2, 4 and 6 wt % of nanoTiC$_p$.

The first stage of powder metallurgy is the analysis of powders. Aluminum powder with a 30–50 μm size range was used in this study. Nano TiC$_p$ particles, which vary from 50 to 80 nm in size, were analyzed individually for further study. Figure 1 shows the Transmission electron micrograph (TEM) of nanoTiC$_p$. The TEM image confirms the specified nanoTiC$_p$ particulates size range.

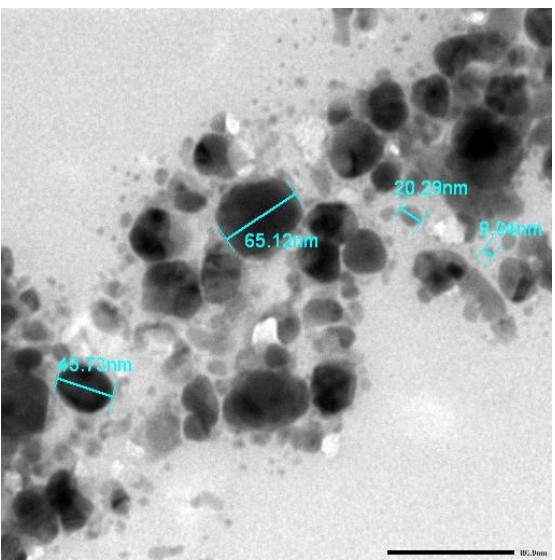

**Figure 1.** TEM image of nanoTiC$_p$.

The second stage of powder metallurgy is the mixing of the powder. The Al and nanoTiC$_p$ powders were weighed independently in different plastic containers. After weighing 2, 4 and 6 weight.% nanoTiC$_p$ particles, they were mixed with aluminum powder and the composite powder was prepared carefully. A mortar and pestle were used for the proper mixing of both powders for half an hour.

The third stage is mechanical alloying. The purpose of mechanical alloying is mixing at the atomic and structural level. Al nanoTiC$_p$ powder mixtures were carried to horizontal ball-milling. The ball mill was designed for milling a total powder charge of 0.50 kg per run. The following are the specifications of the horizontal ball mill: outer diameter = 300 mm, width = 105 mm and rotation speed = 100 rpm.

The stainless-steel balls were used as milling media inside the ball mill and also 2.0 wt % of ethyl acetate was used as a process control agent to diminish the extreme tendency of aluminum to become self-welded during milling. An inert gas such as argon was used while milling to ensure that atmospheric air did not affect the milling process. The ball milling was performed with 100 rpm for 8 h. After alloying, the powder temperature was ~100–120 °C. Hence, the powder was permitted to cool for 4 h inside the ball mill [16]. Then, it was removed from the ball mill and the powder mixture was stored or kept in a plastic container.

Mold cavity is the fourth stage of powder metallurgy. Figure 2a–c show a punch, die and ejector of stainless steel fabricated for powder compaction [11]. The die was greased to ensure the easy removal of the powder pellets. The Al nanoTiC$_p$ composite powders of a weighed amount were mixed with 2 wt % ethyl acetate and poured into the die. The punch was introduced from the top and pressing was performed using an arbor press. The powder compact was ejected out of the die and put in a flexible mold and its mouth was tied tightly with the help of a string. The flexible mold should be of an appropriate size in order to avoid wrinkles on the mold surface. The rubber balloons were used as flexible molds. The flexible mold should be 100 percent leak proof to prevent the leakage of oil into the powder sample during isostatic compaction. This procedure is adopted to fabricate 2, 4 and 6 wt % of nanoTiC$_p$ particulates during mold cavity.

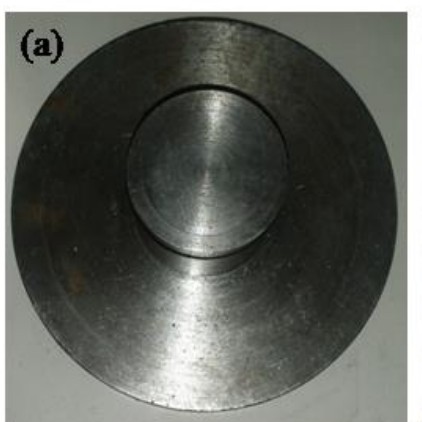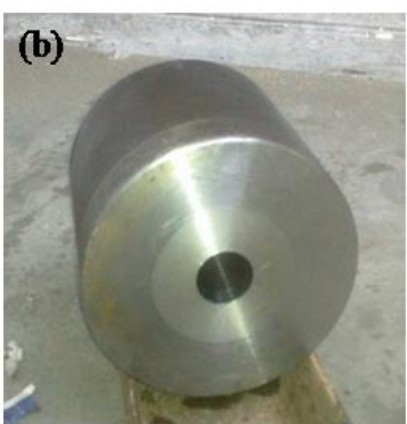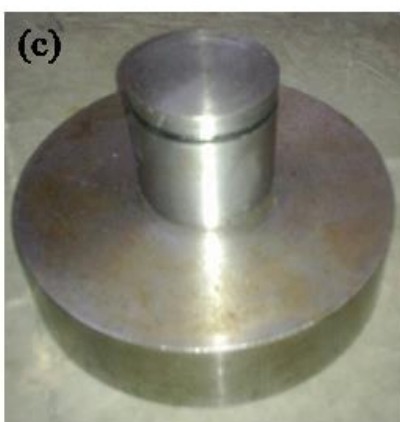

**Figure 2.** (**a**) Image of Punch, (**b**) Image of Die and (**c**) Image of Ejector.

The cold Isostatic compaction process (CIP) is the fifth stage. CIP is a compaction route in which an equal and uniform pressure is applied concurrently from all directions at room temperature, in oil as the medium with powder in a flexible mold to ensure that it can be compacted in the required size and shape. Often, a compacted sample is called a green compact. By this compaction process, it is observed that the theoretical density of the weighed powder parts is found to be approximately 90 to 98%. For the pressurizing medium, water or oil can be applied. Figure 3 shows a compaction process using a cold isostatic compaction chamber.

The final stage is the sintering or annealing of the compacted product. The cold isostatically compacted product was annealed in an inert gas environment (such as Argon). A programmable furnace is used for this purpose. Its temperature was gradually raised up to 600 °C (ramp rate—85 °C/h) and the compacted samples were sintered at 600 °C for about 3 h and 45 min. Again, the specimens were cooled in the furnace for 3 to 4 h (cooling rate—90 °C/h).

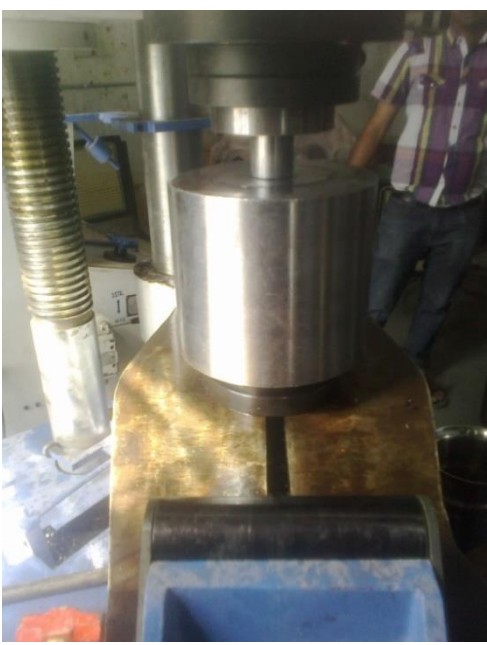

**Figure 3.** Compaction process on 1000 kN UTM using a cold isostatic compaction chamber.

## 4. Investigating of Properties

### 4.1. Micro-Hardness

A digital Vickers micro hardness (VHL-VMHT (Walter UHL) company tester was used for determining the micro-hardness of the composites' material. A diamond indenter of a square pyramid shape was utilized to find the micro-hardness on the surface of the composite material. The impression area is measured by $d^2/2Sin\ (136°/2) = d^2/1854.4$, measuring d (μm) and F (N), the Vickers micro-hardness, Hv, is determined by the following:

$$Hv = 1854.4F/d^2\ (GPa) \tag{1}$$

### 4.2. Porosity

The Archimedes principle was used for finding out the porosity of the annealed as well as the unannealed Al nanoTiC$_p$ composites product. The compacts were first weighed in air and then tied with string and weighed while hanging in water. The following formula is employed to determine the density of the compact product:

$$\rho_s = (w_a \times \rho_w)/(w_a - w_w) \tag{2}$$

where,

$\rho_s$ = Annealed compact product Density (g/cm$^3$);
$\rho_w$ = Water Density (g/cm$^3$);
$w_a$ = Weight of specimen in air (g);
$w_w$ = Weight of specimen in water (g).

The following formula was employed to calculate the porosity of the composite specimen:

$$E = 1 - \rho_s/\rho_t \tag{3}$$

where

E = porosity;
$\rho_s$ = Annealed part density (g/cm$^3$);
$\rho_t$ = Theoretical density (g/cm$^3$).

The theoretical density of the Al nanoTiC$_p$ composites was found using the densities of TiC$_p$ and pure aluminum as follows:

Density of TiC$_p$ = 4.93 g/cm$^3$

Aluminium Density = 2.7 g/cm$^3$

### 4.3. Compressive Strength

The L/D ratio of 1.5 was maintained in the compression test [16] of the Al nanoTiC$_p$ composite specimens with the help of a Computerized UTM machine(Fine Manufacturing Industries, Pune, India) of 1000 kN capacity. The required sized samples were placed between two flat platens and compressed properly and observed maximum recorded load.

### 4.4. Indirect Tensile Strength

The right circular cylindrical shape of Al with 2, 4 and 6 wt % of the nanoTiC$_p$ composites samples were fabricated. To find the indirect tensile strength of the samples, a 1000 kN UTM was used. In this test a right circular cylinder is compacted absolutely along the diameter of the specimen flanked by two flat-plates. Equation four is employed to calculate indirect tensile strength.

$$G = 2F/\pi.d.t \tag{4}$$

where

F = Applied Force (N) ($0 \leq F \leq 1000$ kN);
d = Specimendiameter (m);
t = Specimenthickness (m);
G = Indirect tensile strength (N/mm$^2$).

### 4.5. Microstructural Analysis

The microstructures of the Al nanoTiC$_p$ composites were examined using (Jeol Japan, Model No. JSM-6390A) SEM. The SEM resolution is 30 kV (3.0 nm), and the accelerating voltage is 0.5 kV to 30 kV. The required sized samples were fabricated from the machining (Lathe and Milling). After machining, polishing paper was used for creating a smooth surface. For obtaining a fine structure, the samples were polished on cloth using a disc polisher with diamond-lapping paste for about 40 min to ensure that a mirror-finish was acquired. For etching, 1% Keller Reagent was used for 30 s and then the samples were cleaned with distilled water and the surface of the samples was dried with dry cloths. Then, the SEM images were taken of Al 2, 4 and 6 wt % of nanoTiC$_p$ and studied for micro structural analysis.

## 5. Results and Discussion

The study of mechanical properties such as Vickers micro-hardness, density, porosity, compressive strength and indirect tensile strength and SEM micrograph of Al nanoTiC$_p$ composites were investigated.

### 5.1. Vickers Micro-Hardness

Indentation is performed at 500 gm on each sample at least 10 times. The average value of Vickers micro-hardness is plotted in Figure 4. It is observed from Figure 4 that micro-hardness is increased 33.3% compared to the pure Al powder sample and also the Vickers micro-hardness of the Al 6.0 wt % nanoTiC$_p$ composites is lesser than the Al 4.0 wt % nanoTiC$_p$ composites. The micro-hardness reduces at 6.0 wt % because nanoTiC$_p$ particles are clustered with each other at higher weight percent, due to which the number of effective nanoparticles present is lower [17,18].

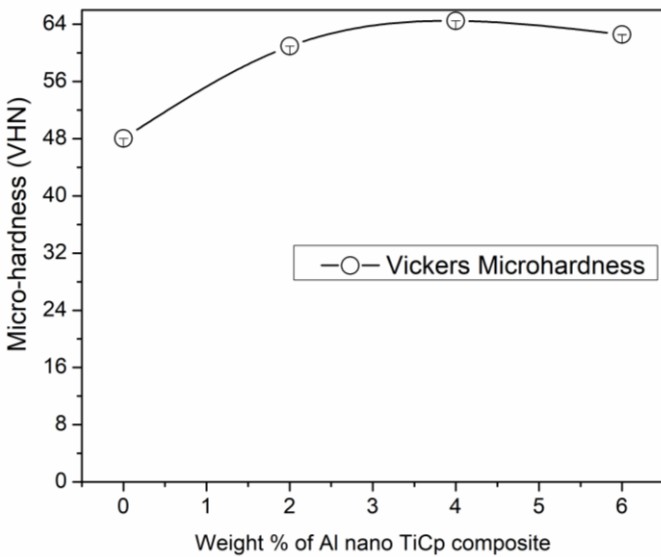

**Figure 4.** Vickers micro-hardness of Al nanoTiC$_p$ composites.

### 5.2. Density and Porosity

The density and porosity of the Al nanoTiC$_p$ composites were found before and after the annealing. Figure 5a–c reveals the information about the actual density, theoretical density and porosity of the Al nanoTiC$_p$ composite before and after sintering. These figures depict clear information about the enhancement of the density with the enrich wt % of nanoTiC$_p$ due to the lower density of Aluminum compared to TiC$_p$. The comparison of Figure 5a,b shows that the sintering causes a reduction in the density of composites product such as the Al nanoTiC$_p$ composites. The reason is that during annealing, ethyl acetate or other volatile mixtures and moisture, etc., are vaporized and, hence, an enhancement of the porosity is observed in Figure 5c and a reduction in the density also, due to the mitigation of residual compressive stresses that were generated during the time of compaction.

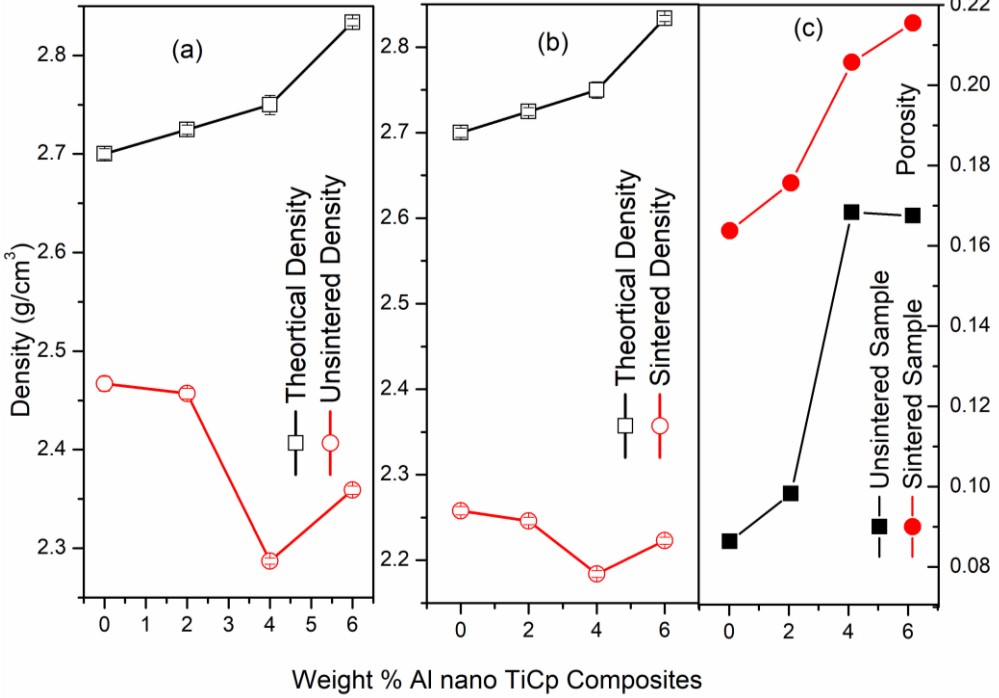

**Figure 5.** (**a**) Density of un-sintered, (**b**) Density of sintered Al nanoTiC$_p$ Composites and (**c**) Porosity of Al nanoTiC$_p$ Composites in sintered and un-sintered conditions.

### 5.3. Compressive Strength

For determining the compressive strength of Al nanoTiC$_p$ composites, a 1000 kN universal testing machine was used. Figure 6 reveals the information about compressive strength vs. wt % of nanoTiC$_p$ composites. Without the addition of nanoTiC$_p$, the aluminum sample's compressive strength is 100 MPa. Nevertheless, for all the nanoTiC$_p$ samples, the compressive strength varied from 450 to 510 MPa. At first, the compressive strength increases with the wt % of TiC$_p$. The increase in compressive strength with wt % of nanoTiC$_p$ is due to the increase in the work hardening rate with a higher wt % of nano particles and the hard nanoTiC$_p$ particles act as an obstacle to the movement of dislocation. It is also observed from Figure 6 that the Al 4.0 wt % nanoTiC$_p$ composites have shown more compressive strength compared to the 6% weight due to the agglomeration of very fine nanoTiC$_p$ particles at a higher wt %. Therefore, the tendency of the effective availability of nano particles reduced, which again diminishes the particle strengthening at a higher wt %.

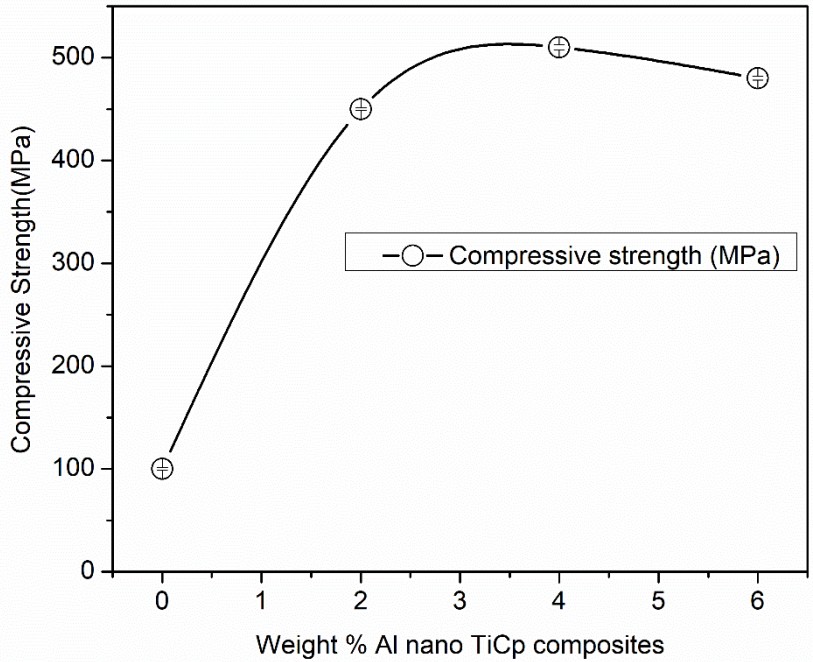

**Figure 6.** Al nanoTiC$_p$ composites compressive strength.

### 5.4. Indirect Tensile Strength

The indirect tensile strength was tested on a 1000 kN UTM machine. The indirect tensile strength of the Al powder sample is 90 MPa. In the nanoTiC$_p$ samples, the indirect tensile strength value varied from 250 to 330 MPa. Again, too much change is observed in the nanoTiC$_p$ samples compared to the pure aluminum powder sample. It was also observed that the indirect tensile strength diminished for the Al 6.0 wt % nanoTiC$_p$ composites compared to the Al 4.0 wt % nanoTiC$_p$ composites in Figure 7. The main cause of this is the agglomeration of very fine nanoTiC$_p$ particles at a higher weight percentage in the Al metal matrix. This further creates large clusters of nano particles. This clustering of nano particles is the main reason of generating stress concentration. After that, crack propagation is started in the grain boundary of the nano materials, which reduces the strength. There are several factors that may cause the influence on the strength of nano composites such as work hardening and deformation, and an enhancement of the dislocation density due to the thermal extension of aluminum and TiC$_p$ particles. This dislocation density affects the grain refinement and the development of sub-grains [17–19].

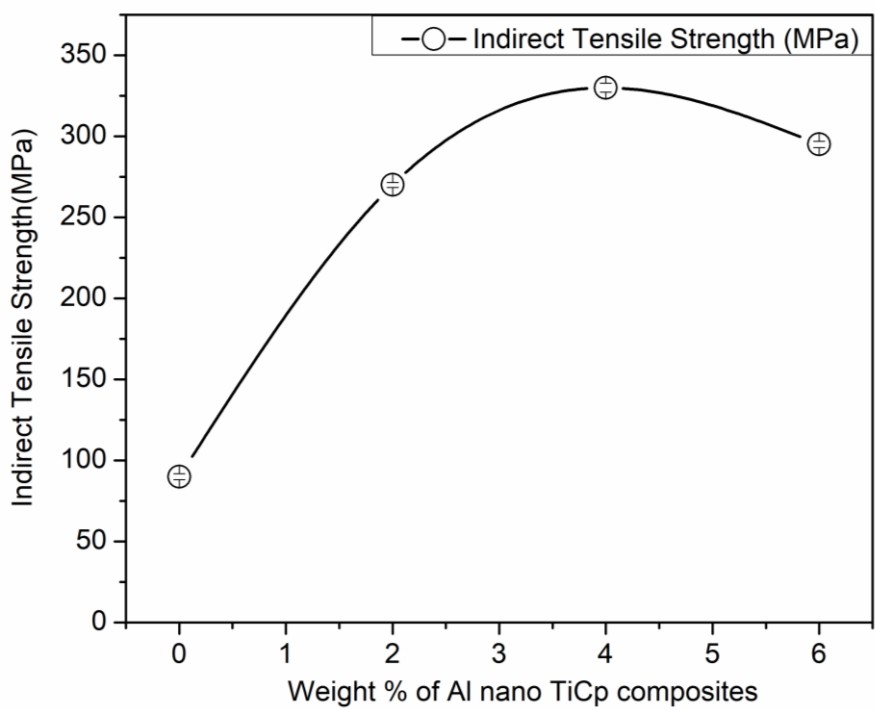

**Figure 7.** Indirect tensile strength of sintered Al nanoTiC$_p$ composites.

*5.5. Microstructural Analysis Using SEM*

The last factor is the nanoTiC$_p$ particles' distribution in the Al metal matrix. To check the particles' distribution, SEM is a great tool. Therefore, micro-structural analysis was conducted. The SEM image of the Al 2 wt % nanoTiC$_p$ composite is shown in Figure 8a. The nanoparticles are uniformly distributed and bonded with the Al matrix. Figure 8b depicts that the TiC$_p$ nano particles are properly fettered to the Al metal matrix; nevertheless, the nanoparticles in the Al 4 wt % TiC$_p$ are greater than in the Al 2 wt % TiC$_p$ composites. In the 4 wt % nanoTiC$_p$, the SEM images show that the TiC$_p$ nanoparticles are consistently dispersed in the aluminum matrix. Too much agglomeration was found at some regions in the Al 6 wt % nanoTiC$_p$, as shown in Figure 8c. That simply indicates the clustering of TiC$_p$ nanoparticles in the Al metal matrix.

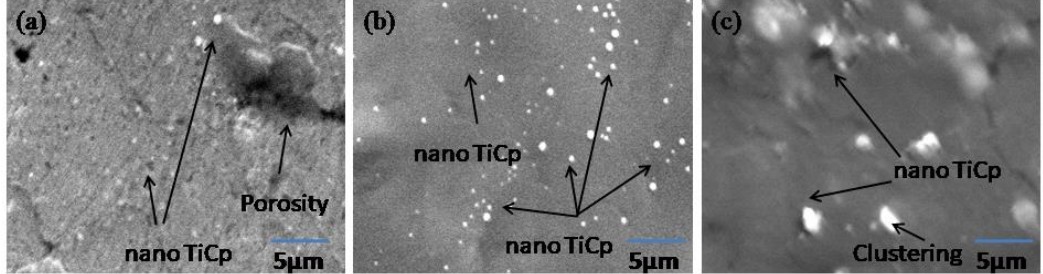

**Figure 8.** SEM Image of (**a**) Al 2 wt %, (**b**) Al 4 wt % and (**c**) Al 6 wt % nanoTiC$_p$ composite after annealing.

## 6. Conclusions

Al nanoTiC$_p$ composites were successfully made with the help of powder metallurgy, including the cold isostatic compaction process at 650 °C with an inert gas atmosphere. A fine, consistent and equiaxed structure of the nano composite powder was obtained by the mechanical alloying of Al and nanoTiC$_p$ powders for 8 h of ball-milling. It is found that the initial manual compact should be made carefully to ensure that the final compact product quality can be improved. The mechanical properties, such as hardness, indirect tensile strength and compressive strength, of the Al nanoTiC$_p$ composites surged up to

4 wt % nanoTiC$_p$ and then declined at 6 wt % of nanoTiC$_p$ due to clustering on the nano particles at a higher wt %. The SEM images show that Al nanoTiC$_p$ composites have a uniform dispersion of nanoTiC$_p$ in the Al matrix up to 4 wt % of nanoTiC$_p$; after 4 wt %, some clustering is observed in the microstructure, which diminishes the overall mechanical properties of the Al nanoTiC$_p$ composites. This clustering is the main cause of the shrinking availability of effective nano particles and, consequently, the strength and hardness are deteriorated after 4 wt % nanoTiC$_p$. Hence, the fabrication of Al nanoTiC$_p$ composites through a powder metallurgy process with a cold isostatic compaction chamber is suitable for telecommunication equipment, lawn and garden equipment, farm/off-road equipment, power and hand tools, sporting goods and firearms in the industrial segment.

**Author Contributions:** G.B. and A.T. carried out the experimental work and the corresponding data interpretation. R.P. and K.S. suggested and guided this research and G.B. wrote the paper. R.D. and V.P. have helped with the interpretation of results and written work. All authors have read and agreed to the published version of the manuscript.

**Funding:** There is no funding for this research.

**Conflicts of Interest:** The authors declare no conflict of interest.

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
