# Peer review of "Development, Testing and Characterization of Al NanoTiCp Composites through Powder Metallurgy Techniques"

_jcs, doi:10.3390/jcs5080224_

Round 1

Reviewer 1 Report

I believe that manuscript requires a lot of work; there is no in-depth interpretation of the results; the benefits obtained by modifying pure Al with nanoparticles are not visible. The presentation of materials and methods should also be refined.

Here are my comments:

  1. You have to decide how to write the words with the nano prefix: Nanotechnology, nano composites or nano-particles.
  2. p.1, l.41: Abbreviations should be clarified the first time they are used in the text. Moreover, throughout the text you write TiC and once TiCp – decide which form is correct.
  3. p.2, l.52: You should enter the Materials section, in which you would list all the starting materials, solvents, etc. and their manufacturers.
  4. p.2, l.55: The subsection numbering is confusing. After all, in 2.1 you do not describe the entire method of obtaining an alloy, but only give preliminary information about its composition, whereas the description of the method is later (in 2.3 and 2.4).
  5. p.2, l.78: What temperature was used?
  6. p.2, l.79: Ethyl acetate was used to reduce the tendency of aluminium to what?
  7. p.3, l.101: Why are sections 2.5 and 2.6 described separately? Are they not about the same process?
  8. p.4, l.127: What was a producer of tester?
  9. Please delete the ellipsis in all equations.
  10. p.4 and 6, Figure 4 and 5: A photo of the apparatus is not needed. Just enter the name, model number and manufacturer.
  11. p.5, l.176: What was a producer of machine?
  12. p.6, l.190: Should be: Applied Force
  13. p.6. l.196: What model of SEM was used, what producer?
  14. p.6: Why was the micro-hardness for alloy Al-4%Nano TiC higher than for alloy with 2% of TiC? What was the hardness of ordinary Al without the addition of nanoparticles? What samples were investigated - annealed or not?
  15. p.8, Figure 7: What was the reason of such high difference between value of theoretical density and density of samples? And the same for porosity. What were the density and porosity for pure Al?
  1. p.8, Figure 8: Is it Porosity in %, as it has been shown in equation 3?
  2. p.9, l.243-248: Strange explanation, because 2% is less and 6% is more than 4%. Therefore, the tendency to agglomeration increases compressive strength or decreases? What was the compressive strength for pure Al?
  1. p.9, l.252: The samples were annealed or not? What was the indirect tensile strength for pure Al?
  2. p.9, l.261: What literature are you referring to here?
  3. p.10, Figure 10: It would be good if you mark in the images, e.g. with arrows, nanoparticles and the matrix.
  4. p.10, l.278: What does FESEM mean?
  5. p.10, l.282: What is greater in 4 wt% than in 2 wt%?
  6. p.10, l.292: Conclusion is not the place to enter shortcuts.

Also English needs to be corrected, such as:

- p.3, l.88: Should be: shows

- p.7, l.224-226: Wrong sentence construction.

- p.7, l.227: Should be: shows

- p.10, l.296-299: The construction of the sentence is wrong.

Reviewer 2 Report

This could be an interesting investigation, but the paper lacks the rigour and the depth of a scientific article. Makor revisions are needed to get it to a publishable level.

Overall:

English needs to be thoroughly revised. Please decide if you want to use American or British English and stick with it.

The formatting needs more attention.

Please define the acronyms the first time you use them.

Abstract:

This section contains too many generic sentences, too unprecise and unreferenced to be worth of a research article. Please revise it fully, both in terms of technical content and writing style.

“The mechanical properties like density, porosity, mi-18 cro-hardness; compressive strength & indirect tensile strength were enhanced up to Al-4 wt% of 19 nano TiCp composites.” This sentence is very unclear.

“it raises several properties like chemical, mechanical, optical and/or physical 28 properties in different aspects” please clarify, this sentence is too generic and unreferenced to be meaningful.

Fabrication of Al-Nano TiC using Powder Metallurgy

This subsection is too vague, modified in what sense? what machine did you use? What process parameters?

Analysis of Particles

“The TEM image con-61 firms the specified nano TiC particulates size range.” Does it? Did you conduct a statistical analysis over a wider sample or are you basing your conclusion only on a single image?

Powder Mixing

Does the “mortar and pestle” mixing have a effect on the powder particle size?

Mechanical alloying

Please remove the “:” from the subsection title.

You need to provide more details about the ball-milling machine you used.

“…was 79 used as process-control-agent to diminish extreme tendency of aluminum.” Tendency to do what?

“…and reserved in the container”, what do you mean?

Cavity of Mould

Please remove the “:” from the subsection title.

What do you mean for “Gum”?

Overall, the manufacturing technique is poorly explained, and hardly understandable for a non-specialist in powder metallurgy. No process parameters are provided making it impossible to reproduce the obtained results.

Cold Isostatic Compaction

This process is described in a very poor and undetailed way. You need to provide process parameters and all the details of the used equipment.

Annealing of Al-nano TiC Compacts:

Please remove the “:” from the subsection title.

Again, no details regarding ramp-rate and cooling rate are provided.

Micro-Hardness

Again, the most important parameters, i.e. the dimension and the exact geometry of the indenter, are not provided.

Porosity

“Weighing machine”, do you mean a scale?

“…after that tied with thread and weighed whilst lynching in water”, WHAT???

Equation 4 is useless.

Compressive Strength

“The L/D ratio of 1.5 were maintained in the Compression test”, no idea what are you talking about.

Are you sure the machine is 1000 kN? By the way it is kN not KN.

Indirect Tensile Strength

This test is described very very poorly.

Microstructural Analysis

Again, too superficial.

Vickers micro-hardness

Without standard deviation, the obtained results shown in figure 6 do not mean anything, as it is not possible to judge if there is a statistical difference between the 3 sets of results. The Y axis of the figure should start from 0.

“% Nano TiC Composites. Hardness reduces at 6.0 wt% because nanoTiCp particles are clustered with each other at higher weight percentage [17].” This sentence is a mess. Do you have micrographies to support this?

Porosity

It seems to me that in the discussion you are confusing density and porosity.

Compressive strength

How many specimens did you test? How were the data post-processed? I do not believe you have such a tight standard deviation. Y axis of figure 8 is misleading.

Why do you have that behaviour in your tests? The dependency of the strength on W% is not explained.

Indirect tensile strength

How many specimens did you test? How were the data post-processed? I do not believe you have such a tight standard deviation. Y axis of figure 9 is misleading.

Why do you have that behaviour in your tests? The dependency of the strength on W% is demonstrated by images of the materials.

Microstructural Analysis using SEM

Those images presented here and in this way do not have any meaning.

Conclusions

This section is poor and should be further enriched explaining why these results are relevant on an industrial point of view and what the next steps will be.

Round 2

Reviewer 1 Report

I would like to thank the authors for their explanations and taking into account my comments. I am sorry that my reply took so long. I believe that the research in the manuscript is now clearly described and the results are well interpreted. After considering a few minor remarks, the paper can be published.

  1. p.1, l.34: Should be: nano particles
  2. p.1, l.38: TiC is a different compound than your TiCp?
  3. p.2, l.54: Should be: 30-50 μm
  4. p.3, l.84: I cannot see where is the added temperature of milling?
  5. p.4, l.139: After all, this Figure 4 is not there! Instead, the manufacturer and model of the tester should be given.
  6. p.5, l.154-157, 169-170: You mean: g/cm3 and g?
  7. p.5, l.174: What was the producer of Computerized-UTM machine?

Author Response

Please have a look at attachment of word file.

Reviewer 2 Report

The comments have been addressed properly

Author Response

Thanks for this acceptance information.

Round 3

Reviewer 1 Report

In my opinion, the current version is complete and can be published in JCS. I am asking only to change "30-50micron" to "30-50 µm" (p.2, l.55).